# Deep ensembles based on Stochastic Activation Selection for Polyp Segmentation

**Alessandra Lumini**[1]                                                    ALESSANDRA.LUMINI@UNIBO.IT
**Loris Nanni**[2]                                                              LORIS.NANNI@UNIPD.IT
**Gianluca Maguolo**[2]                                          GIANLUCA.MAGUOLO@PHD.UNIPD.IT
[1] *DISI, Università di Bologna, Via dell'università 50, 47521 Cesena, Italy*
[2] *DEI, University of Padua, viale Gradenigo 6, Padua, Italy*

**Editors:** Under Review for MIDL 2021

## Abstract

This work deals with accurate polyp detection and segmentation. We compare some variants of the DeepLab architectures obtained by varying the encoder backbone. We compare several encoder architectures obtained by substituting ReLU activation layers with other functions. Our experimental evaluations show that our best ensemble produces very good segmentation results by achieving high evaluation scores with a dice coefficient of 0.884 for the Kvasir-SEG dataset. The MATLAB source code is available at GitHub:
https://github.com/LorisNanni/Deep-ensembles-based-on-Stochastic-Activation-Selection-for-Polyp-Segmentation.

**Keywords:** segmentation, convolutional neural network, colonoscopy, deep ensembles.

## 1. Introduction

Colorectal Cancer is one of the prominent causes of cancer related deaths worldwide (Jha et al., 2020). Polyps are predecessors to this type of cancers and therefore important to find and accurately segment them through colonoscopy examinations. However, accurate polyp segmentation is a challenging task. The objectives of this paper include: (1) employing some of the most popular deep CNN architectures extensively used in computer vision community for semantic image segmentation; (2) modifying base backbones by substituting ReLU activation layers with other functions; (3) investigating the feasibility of design deep ensembles by fusing perturbed backbones.

## 2. Deep Learning for Semantic Image Segmentation

Our decoder network is DeepLabv3+ (Chen et al., 2018). Apart from the main architecture of the network, there are a handful of other good design choices that would help achieve good performance. For example, the choice of a pretrained backbone for the encoder part of the network. Among several CNNs widely used for transfer learning we tested the following backbones: ResNet18 and ResNet50. Also the choice of the loss function influences the way the network is trained. The dice loss is used in this work. Moreover, the choice of the activation function can be significant. ReLU is the nonlinearity that most works use in the area, but several works have reported improved results with different activation functions. Finally, we perform experiments with data augmentation, consisting in horizontal and vertical flips and rotations of 90°.

## 3. Stochastic Activation Selection

This method was first introduced in (Nanni et al., 2020). The process to create a new network is based on the replacement of each activation layer (ReLU) by a new activation function which can be fixed a priori or randomly selected from the ones in the pool. Since this is a random procedure, it yields a different network every time. We train each network independently on the same set of data and then we merge their results using the sum rule. The pool of activation functions is made by ReLU and a list of its modifications proposed in the literature: ReLU, Leaky ReLU, ELU, PReLU, S-Shaped ReLU (SReLU), Adaptive Piecewise Linear Unit (APLU), Mexican ReLU (MeLU) (with $k = 4, 8$), Gaussian Linear Unit (GaLU) (with $k = 2, 4$), PDELU, Swish (fixed and learnable), Soft Root Sign, Mish (fixed and learnable) and Soft Learnable. See (Nanni et al., 2020) for details on these activation functions.

## 4. Experiments and discussion

All the experiments have been carried out on the Kvasir-SEG dataset (Jha et al., 2020) which includes 1000 polyp images. For a fair comparison with other approaches (see table 4) as (Jha et al., 2021) and (Huang et al., 2021) we use the following testing protocol: 880 images are used for training, and the remaining 120 for testing. The first experiment (Table 1) is aimed at comparing the different backbone networks listed in section 2 and at designing effective ensembles by varying the activation functions. Each ensemble is fusion by the sum rule of 14 backbones (since we use 14 activation functions). The ensemble name is the concatenation of the name of the backbone network and a string to identify the creation approach. These strings are **act**, **sto**, **relu**. In **act**, each network is obtained by deterministically substituting each activation layer by one of the activation functions of section 3 (the same function for all the layers, but a different function for each network). **sto** is an ensemble of stochastic models, whose activation layers have been replaced by a randomly selected activation function (which may be different for each layer). **relu** is an ensemble of original backbones which differ only for the random initialization before training; it means that all the starting models in the ensemble are the same, except for the initialization.

Table 1: Experiments with different backbones

| Method | IoU | Dice | F2 | Prec. | Rec. | Acc. |
|---|---|---|---|---|---|---|
| resnet18 | 0.759 | 0.844 | 0.845 | 0.882 | 0.856 | 0.952 |
| resnet50 | 0.751 | 0.837 | 0.836 | 0.883 | 0.845 | 0.952 |
| resnet18-352 | 0.787 | 0.865 | 0.871 | 0.891 | 0.884 | 0.960 |
| resnet50-352 | 0.801 | 0.872 | 0.884 | 0.881 | 0.900 | 0.964 |
| resnet50act | 0.779 | 0.858 | 0.859 | 0.894 | 0.869 | 0.957 |
| resnet50relu | 0.772 | 0.855 | 0.858 | 0.889 | 0.870 | 0.955 |
| resnet50sto | 0.779 | 0.859 | 0.864 | 0.891 | 0.877 | 0.957 |
| resnet50-352sto | 0.820 | 0.885 | 0.888 | 0.915 | 0.896 | 0.966 |

Clearly, using larger input sizes boosts the performance of Resnet50 (see Table 1); Unless otherwise specified, the size of the input image is 224x224, however it might be different in some networks. In that case, it is reported after the name of the backbone CNN. The best performance, among the ensembles, is obtained by resnet50-352sto.

A comparison with some state-of-the-art results is reported in Table 2. The results of many methods that we compare with are reported by (Jha et al., 2021), please read it for the original reference of a given approach.

Table 2: State-of-the-art approaches

| Method | IoU | Dice | F2 | Prec. | Rec. | Acc. |
|---|---|---|---|---|---|---|
| resnet50-352sto | 0.820 | 0.885 | 0.888 | 0.915 | 0.896 | 0.966 |
| U-Net | 0.471 | 0.597 | 0.598 | 0.672 | 0.617 | 0.894 |
| ResUNet | 0.572 | 0.69 | 0.699 | 0.745 | 0.725 | 0.917 |
| ResUNet++ | 0.613 | 0.714 | 0.72 | 0.784 | 0.742 | 0.917 |
| FCN8 | 0.737 | 0.831 | 0.825 | 0.882 | 0.835 | 0.952 |
| HRNet | 0.759 | 0.845 | 0.847 | 0.878 | 0.859 | 0.952 |
| DoubleUNet | 0.733 | 0.813 | 0.82 | 0.861 | 0.84 | 0.949 |
| PSPNet | 0.744 | 0.841 | 0.831 | 0.89 | 0.836 | 0.953 |
| DeepLabv3+ ResNet50 | 0.776 | 0.857 | 0.855 | 0.891 | 0.861 | 0.961 |
| DeepLabv3+ ResNet101 | 0.786 | 0.864 | 0.857 | 0.906 | 0.859 | 0.961 |
| U-Net ResNet34 | 0.81 | 0.876 | 0.862 | 0.944 | 0.86 | 0.968 |
| HarDNet-MSEG | 0.848 | 0.904 | 0.915 | 0.907 | 0.923 | 0.969 |

Our best approach obtains best performance but HarDNet-MSEG. However, this is might due to the fact that their data augmentation is larger than ours.

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
