# OpenReview forum: "Deep ensembles based on Stochastic Activation Selection for Polyp Segmentation"
_MIDL.io/2021/Conference/Short — MIDL 2021 Poster_

### Official Review · Reviewer_1GDN · 2021-04-23

**Confidence:** 5
**Final Rating:** 4

**Summary:**

This papers applies a method for stocastic selection of activation functions in the backbone of ensembles of DeepLab v3+ segmentation encoder-decoder architectures. The stocastic selection method is already published by the authors, however applied to different data on this work. Using the stocastic selection in building and ensemble improved the results of a ResNet 50 backbone.



**Strengths:**

The proposed stochastic activation resnet outperformed other widely used segmentation architectures, although it did not achieve state-of-the-art on the used dataset. The interesting part of this manuscript is the improvement in results not only from using ensembles, but ensembles of perturbed architectures in their activation function, a concept i personally had not seen before.

Code and data are public.

**Weaknesses:**

The manuscript in general calls the backbone "model", but the model includes the DeepLab V3+ architecture after the backbone. The text should be more clear that the study is over backbone peturbation.


**Deanonymize Review:**

yes

**Detailed Comments:**

In Table 1, I think you should make it clear that the variable is not the whole model, but the DeepLab v3+ backbone.

In the end of page 2 and beginning of page 3, the same sentence is repeated: "the
best performance, among the ensembles, is obtained by resnet50-352sto."

The last sentence/conclusion about the method not achieving state-of-the-art performance because of HarDNet-MSEG could be rewritten.

The github link should link directly to the relevant repository, not to the user.

**Justification Of The Rating:**

Besides minor improvements that i described above, the idea of building ensembles by pertubation of activation functions is very interesting and would be of interest for a poster presentation for the community.

**Paper Type:**

validation/application paper

**Special Issue:**

no

---

### Official Review · Reviewer_qFXp · 2021-04-26

**Confidence:** 3
**Final Rating:** 3

**Summary:**

The authors propose to modify DeepLab architecture with ResNet18 or ResNet50 as the backbone network. They propose ensembling models based on Stochastic Activation Selection. Experiment results on the Kvasir-SEG dataset show that their proposed resnet50-352sto ensemble approach can outperform several state-of-the-art approaches.

**Strengths:**

1. The idea of utilizing Stochastic Activation Selection for model ensembles seems to be novel.
2. The authors conduct many experiments for locating the best ensemble approach. The authors also compare their method with the state-of-the-art results to justify their method.

**Weaknesses:**

1. The authors' implementations seem to have problems. The authors' ensemble models approach has a similar performance comparing to the single model approach reported by others.

As shown in Table 2 line 9, DeepLabv3+ ResNet50 has an IoU score of 0.776 for the Kvasir-SEG dataset in Jha’s paper [1]. As shown in Table 1 line 2, DeepLabv3+ ResNet50 implemented by the authors has an IoU score of 0.751 (0.025 worse than the result reported in Jha’s paper). As shown in Table 1 line 6, DeepLabv3+ ResNet50 ensemble models have an IoU score of 0.772 (0.004 worse than the result reported in Jha’s paper).

[1] Jha, Debesh, et al. "Real-time polyp detection, localization and segmentation in colonoscopy using deep learning." Ieee Access 9 (2021): 40496-40510.

2. Performance improvement seems mostly due to the large input size.

As shown in Table 1 line 5, line 6, and line 7, different activation choices do not affect the performance a lot. As shown in Table 1 line 3, line 4, and line 8, when authors utilize large input size, the results improve a lot.

As the authors mentioned in the paper: "using larger input sizes boosts the performance." However, details about the input size information are missing in Table 2.

3. Some parts of the paper need further clarification. Please check on my detailed comments.


**Deanonymize Review:**

yes

**Detailed Comments:**

Some parts of the paper need further clarification:

Example 1:

In the Abstract section, the authors mention that: "We compare some variant of the DeepLab architecture obtained by varying the **decoder backbone**. We compare several **decoder architecture** and we perturb their layers by substituting ReLU activation..."

In the Deep Learning for Semantic Image Segmentation section, the authors mention that: "For example, the choice of a pretrained backbone for the **encoder part of the network**…"

My questions are:

(1) Is this a typo? (in bold)  Seems that the decoder backbone is not changed in the work.

Typo here:

variant->variants

several decoder architecture->several decoder architectures

(2) What are the different decoder architectures compared in this work? Seems that the decoder architecture the author used is from DeepLabv3+.

(3) Are ReLU activation layers substituted on both the encoder and the decoder or only on the decoder? In the Abstract section, it seems it is applied to the decoder only. In Sections 3 and 4, it seems that it is applied to both the encoder and the decoder.

Example 2:

The authors should explain that the 352 in “resnet50-352” is the input size in the paper.

After I read the authors' long version of this paper online, I know 352 means the input size.

Link is here: https://arxiv.org/abs/2104.00850

**Justification Of The Rating:**

The method proposed by the authors has some novelty. Consider it is a short paper focus on a novel idea, I recommend the weak acceptance of the paper. However, multiple parts of the paper need further clarification. I recommend the authors revise the paper if it is accepted.

**Paper Type:**

methodological development

**Special Issue:**

no

---

### Meta-Review · Area_Chair_37mW · 2021-05-07

**Recommendation:** Accept (Poster)
**Confidence:** 5

**Metareview:**

The reviewers mention some issues, mostly regarding the presentation, but generally agree that the paper should be accepted.

---

### Decision · Program_Chairs · 2021-05-11

Accept (Poster)